# Differential Expression of *BOC*, *SPOCK2*, and *GJD3* Is Associated with Brain Metastasis of ER-Negative Breast Cancers

**DOI:** 10.3390/cancers13122982

**Published:** 2021-06-15

**Authors:** Rute M. S. M. Pedrosa, Leonoor V. Wismans, Renata Sinke, Marcel van der Weiden, Casper H. J. van Eijck, Johan M. Kros, Dana A. M. Mustafa

**Affiliations:** 1Department of Pathology, Erasmus University Medical Center, Wytemaweg 80, 3000 DR Rotterdam, The Netherlands; r.pedrosa@erasmusmc.nl (R.M.S.M.P.); m.vanderweiden@erasmusmc.nl (M.v.d.W.); j.m.kros@erasmusmc.nl (J.M.K.); 2The Tumor Immuno-Pathology (TIP) Laboratory, Erasmus University Medical Center, Wytemaweg 80, 3000 DR Rotterdam, The Netherlands; l.wismans@erasmusmc.nl (L.V.W.); c.vaneijck@erasmusmc.nl (C.H.J.v.E.); 3Department of Surgery, Erasmus University Medical Center, Wytemaweg 80, 3000 DR Rotterdam, The Netherlands; 4Laboratory of Pathology, Pathan BV, Kleiweg 500, 3045 PM Rotterdam, The Netherlands; R.sinke@pathan.nl; 5Brain Tumor Center, Erasmus Medical Center, Wytemaweg 80, 3000 DR Rotterdam, The Netherlands

**Keywords:** *BOC*, *GJD3*, *SPOCK2*, ER-breast cancer, brain metastasis, blood-brain barrier

## Abstract

**Simple Summary:**

Brain metastasis is diagnosed in 30–50% of metastatic breast cancer patients with currently limited treatment strategies and usually short survival rates. In the present study, we aim to identify genes specifically associated with the development of brain metastasis in breast cancer. Therefore, we compared RNA expression profiles from two groups of patients with metastatic breast cancer, with and without brain involvement. Three genes *BOC*, *SPOCK2*, and *GJD3* were overexpressed in the group of primary breast cancers which developed brain metastasis. Expression profiles were confirmed in an independent breast cancer cohort for both *BOC* and *SPOCK2*. In addition, differential overexpression of *SPOCK2* and *GJD3* mRNA levels were found to be associated with the development of brain metastasis in an external online database of 204 primary breast cancers. Verification of these genes as biomarkers for brain metastasis development in primary breast cancer is warranted.

**Abstract:**

Background: Brain metastasis is considered one of the major causes of mortality in breast cancer patients. To invade the brain, tumor cells need to pass the blood-brain barrier by mechanisms that are partially understood. In primary ER-negative breast cancers that developed brain metastases, we found that some of the differentially expressed genes play roles in the T cell response. The present study aimed to identify genes involved in the formation of brain metastasis independently from the T cell response. Method: Previously profiled primary breast cancer samples were reanalyzed. Genes that were found to be differentially expressed were confirmed by RT-PCR and by immunohistochemistry using an independent cohort of samples. Results: *BOC, SPOCK2,* and *GJD3* were overexpressed in the primary breast tumors that developed brain metastasis. *BOC* expression was successfully validated at the protein level. *SPOCK2* was validated at both mRNA and protein levels. *SPOCK2* and *GJD3* mRNA overexpression were also found to be associated with cerebral metastasis in an external online database consisting of 204 primary breast cancers. Conclusion: The overexpression of *BOC, SPOCK2*, and *GJD3* is associated with the invasion of breast cancer into the brain. Further studies to determine their specific function and potential value as brain metastasis biomarkers are required.

## 1. Introduction

Breast cancer is one of the most notorious cancers associated with brain metastasis [1]. The appearance of brain metastasis invariably heralds the terminal stage of the disease and therefore, prevention of cerebral metastases would be a major step in improving the outlook of patients suffering from breast cancer [2]. Various parameters such as early-onset breast cancer, human epidermal growth factor receptor 2 (HER-2) upregulation, high tumor grade, and estrogen receptor negativity (ER-) were identified as independent risk factors for the development of brain metastasis [3,4]. The spread of tumor cells to the brain is a complicated process consisting of a series of subsequent and interrelated events. One essential step is the penetration of the blood-brain barrier (BBB) by circulating tumor cells. The molecules involved and pathways used by tumor cells to pass through the BBB remain largely unknown [5]. In a previous study, we identified genes involved in the T cell response to play important roles in the trespassing of the BBB of ER- breast cancer [6]. The aim of the present study was to identify genes, other than those related to the T cell response, that are involved in the formation of brain metastasis of ER- breast cancers. The differences in gene expression between specimens of primary ER- breast cancers that were associated with systemic metastases, with and without brain involvement, were identified. Since we were interested in the genes specifically acting in the formation of brain metastases, we compared only these two groups and did not include primary tumors from patients without systemic metastases. Based on fold-of-changes (FOC) and *p*-values, 55 differentially expressed genes were identified. We validated the highest-ranked genes by RT-PCR and immunohistochemistry in primary ER- breast cancer specimens of patients with metastatic disease, with or without cerebral metastases.

## 2. Materials and Methods

### 2.1. Tissue Sample Selection

All tissue samples included and analyzed in this study were primary breast cancer samples. To identify genes involved in the formation of cerebral metastasis, we reanalyzed a previously generated RNA expression database [6]. Fresh-frozen (FF) samples of primary breast cancers from patients with systemic metastases, with and without brain involvement, were used (*n* = 13 and *n* = 9, respectively). Morphological assessment, RNA expression profiling, and relevant clinical data regarding this discovery cohort were provided previously [6]. Based on similar selection criteria, a validation cohort consisting of 30 FFPE primary ER- breast cancer samples was assembled: ten samples from patients who had developed brain metastasis and 20 samples from patients with metastasized cancer without brain involvement. The relevant clinical data of this cohort is provided in Table 1. This study was approved by the Medical Ethics Committee of the Erasmus Medical Center, Rotterdam, the Netherlands and performed in adherence to the Code of Conduct of the Federation of Medical Scientific Societies in the Netherlands (http://www.fmwv.nl/, accessed date 6 April 2021) (MEC 02·953). 

### 2.2. Morphological Assessment

Hematoxylin and eosin (H&E)-stained 5-µm-thick sections from each sample, prepared before and after sectioning for RNA isolation, were evaluated by an experienced pathologist, whereas a tumor cell area was selected for RNA isolation.

### 2.3. Reverse Transcription-Polymerase Chain Reaction (RT-PCR)

Total RNA from FFPE patient material was extracted from 10–15 sections, each of 5 μm (depending on the size of the sample) using the RNeasy FFPE Micro kit (Qiagen, Hilden, Germany). The quantity and quality of the isolated RNA were assessed by the 2100 Bioanalyzer (Agilent Technologies, Santa Clara, CA, USA). Samples were excluded if the yield did not reach a minimum of 100 ng/μL, with a minimum template size of 150 nucleotides. Reverse transcription was performed using the RevertAid H-Minus first-strand cDNA synthesis kit (Thermo Scientific, Vilnius, Lithuania) according to the manufacturer’s protocol. Quantitative real-time PCR (RT-PCR) was performed using TaqMan Master Mix (Applied Biosystems, Austin, TX, USA) on the 7500 RT-PCR system, v.2.3 (Applied Biosystems, Foster City, CA, USA). The following commercially available exon-spanning TaqMan Gene Expression Assays (Applied Biosystems, Foster City, CA, USA) were used: BOC, exon 4-5 (Hs00264408_m1), GJD3/CX30.2, exon 1-1 (Hs00987388-s1), SPOCK2, exon 4-5 (Hs00360339_m1), HPRT1, exon 2-3 (Hs02800695_m1), and HMBS, exon 13-14 (Hs00609296_g1). HPRT1 and HMBS were used as reference genes. The relative quantification of target gene expression was performed using the 2-ΔΔCt comparative method and the threshold cycle value was defined by the point at which there was a statistically significant detectable increase in fluorescence.

### 2.4. Immunohistochemistry

Anti-BOC (1:1000, bs-12322R, Bio-Connect, Huissen, The Netherlands), Anti-SPOCK2 (1:800, HPA044605, Merck, Darmstadt, Germany), and Anti-GJD3 (1:100, 40-7400, ThermoFisher, Rockford, MI, USA) antibodies were used according to the manufacture instructions. For semi-quantitative determination of protein expression, the protocol of Crowe et al. was followed [7]. The immunostained slides were scanned by a Nanozoomer 2.0HT scanner (40× magnification, Hamamatsu Photonics, Hamamatsu, Japan) and four tumor regions of interest (ROIs) per slide were selected. ROIs were evaluated with a semi-quantitative IHC method [7]. ImageJ Fiji 1.52p software (USA) was used to deconvolute the selected immunostained ROIs and convert the slides into gray shades. The shades are related to the counterstained nuclei and thresholds set accordingly. Final intensity scores are given in units and are based on the mean grey values and area intensity ratios of each specific IHC staining. All four ROIs per slide were implemented in the evaluation.

### 2.5. Statistics

Statistical comparisons of immunostaining results and online mRNA data were performed using the Mann–Whitney U test. RT-PCR data were analyzed using unpaired two-tailed Student’s *t*-test (significance levels *p* < 0.05). Data were analyzed and graphs were made using GraphPad software (GraphPad Prism 5.0, San Diego, CA, USA). 

## 3. Results

### 3.1. Identification of Genes Involved in Brain Metastasis of Primary Breast Cancer

Analysis of the expression arrays revealed that 55 genes were differentially expressed between the group with (BM+) and without (BM−) brain metastasis (*p* < 0.01, Figure 1a,b). Out of the 55 genes, 38 were excluded because they presented a FOC < 1.5. Eight out of 17 differentially expressed genes, with a FOC > 1.5, were associated with the regulation of the immune system and were reported on in a previous study [6]. From the other genes, three were identified with either highest FOC, or most significant difference in expression level: *BOC* (FOC = 2.01; *p* = 0.0041), *SPOCK2* (FOC = 1.91; *p* = 0.006), and *GJD3* (FOC = 1.52; *p* < 0.001). All three genes were upregulated in the BM+ group (Figure 1c).

### 3.2. Validations at mRNA and Protein Level

#### 3.2.1. BOC mRNA Expression and Immunohistochemistry

*BOC* mRNA levels were successfully evaluated in 60% of the BM+ samples and in 60% of the BM− samples. *BOC* mRNA expression levels presented a trend towards higher expression levels in the BM+ group (Figure 2), and this trend was confirmed by data mining in a publicly available online dataset of 204 primary breast cancers (Appendix A). Therefore, mRNA levels of *BOC* were consistent with the RNA expression profiles data.

Immunohistochemistry was successfully carried out for 9/10 BM+ and for 20/20 BM− samples. The immunohistochemistry results of *BOC* revealed significantly higher expression in the BM+ group than in the BM− group (median intensity of 24,143 and 21,925 units in BM+ and BM− groups, respectively, *p* = 0.022; Figure 3a). *BOC* expression in both groups revealed a pale homogeneous cytoplasmic staining throughout tumor areas. Roughly 70% of BM+ samples showed higher mean intensity values than those in the BM− group (Figure 3 and Figure 4a,b).

#### 3.2.2. SPOCK2 mRNA Expression and Immunohistochemistry

*SPOCK2* mRNA was significantly overexpressed in the BM+ group (0.21 ± 0.02 vs. 0.32 ± 0.04; *p* = 0.022, Figure 2). The overexpression of *SPOCK2* mRNA in primary breast cancers that metastasized to the brain was further corroborated by data mining in a publicly available online dataset of 204 primary breast cancers (Appendix A, *p* = 0.0026). Herein, the *p* value becomes significant as the group narrows into exclusively brain metastasis (BM + O vs. BM, Appendix A)

The immunohistochemistry results of *SPOCK2* revealed a significant differential expression level between both groups (*p* = 0.02, Figure 3b and Figure 4c,d). The mean intensity in the BM+ group was 47,500 units, with the 25th–75th percentiles ranging over 54,834 units (as compared to a mean of 33,100 units and a 25th–75th percentile-range of 15,750 units in the BM− group (Figure 3b). *SPOCK2* staining displayed considerable heterogeneity in the BM+ tumor areas (Figure 4c).

#### 3.2.3. GJD3 mRNA Expression and Immunohistochemistry

All quality-approved 30 breast cancer samples, measured with different dilutions, delivered a Ct value below the threshold of detection of the RT-PCR method used. Nevertheless, the overexpression of *GJD3* mRNA in primary breast cancers that developed brain metastasis was successfully confirmed in a publicly available online dataset of 204 primary breast cancers, when considering only breast cancers which metastasized exclusively to the brain (Appendix A, *p* = 0.0149). There was a strong heterogeneous cytoplasmic staining, as measured by immunohistochemistry, ranging from strong to weak (Figure 4e,f), with a median difference of 37,599 units between BM+ and BM− groups (108,042 and 70,443 units, respectively). Mean *GJD3* protein expression levels, between the BM+ and BM− groups, were 89,636 vs. 78,332, respectively (FOC = 1.14), with a higher trend towards the BM+ group, though not significant (*p* = 0.06; Figure 3c).

## 4. Discussion

In the current study, we found that the expression of *BOC, SPOCK2,* and *GJD3* is upregulated in the ER- primary breast cancer samples of patients who developed brain metastases.

The immunohistochemistry results for *BOC* confirmed the results of the gene expression profiles: *BOC* was found to be a significantly overexpressed protein in our validation cohort, associated with cerebral metastasis of ER- breast cancer (Figure 3a) and had not been reported previously. *BOC* mRNA levels in the independent cohort (Figure 2) presented a trend towards overexpression in the breast cancers that metastasized to the brain, which was corroborated, in a similar fashion, in the online dataset of 204 primary breast cancers (*p* > 0.05, Appendix A [8]). This lack of significance in the *BOC* mRNA levels between breast cancers with and without brain metastasis might be reasoned by the combination of different factors: a narrow low intensity level difference between both groups (Figure 3a and Figure 4a,b) and a low number of successfully evaluated mRNA samples, from a group of primary breast cancers which lack brain metastases exclusivity (Table 1). *BOC* is a member of the Ig/FNIII repeat family of receptor-like proteins [9]. BOC is an essential component of the Hedgehog (Hh) pathway that promotes Sonic HH (SHH) signaling [10,11]. SHH is essential for the normal development of the nervous system [12] and plays a role in the maintenance and regeneration of adult tissues [13]. All mammalian SHH proteins interact with *BOC* and CDO, another component of SSH signaling [14]. Abnormal SHH signaling is not only associated with developmental defects, but also with cancer [15,16]. Since CDO and *BOC* join the cadherin-β-catenin complexes to contribute to cell signaling at sites of cell–cell adhesion [17], it is tempting to speculate that signaling activities from these complexes are involved in cytoskeletal structural changes, analogous to the formation of protrusions enabled by Cdon in zebrafish neural crest migration [18]. In chick embryos, *BOC* and CDO were found spatially linked in long filopodial extensions that need SHH signaling [19]. The relation between *BOC* expression and metastasis has been described in non-small-cell lung cancer and cancers of the pancreas and prostate. In addition, there are data supporting the contribution of SHH signaling to increased proliferation, invasion, and migration of breast cancer cells [20,21,22]. Inhibition of SHH signaling was shown to reduce these metastatic traits [22,23,24].

*SPOCK2* was significantly upregulated in breast cancer cells that metastasized to the brain. The expression of *SPOCK2* in this context was successfully validated both at mRNA (Figure 2) and protein levels (Figure 3b and Figure 4c). Moreover, we were able to show a strong association of *SPOCK2* expression at the mRNA level in breast cancers with exclusive cerebral metastasis in an independent online cohort of 204 primary breast cancers (Appendix A). *SPOCK2* is a member of the Ca^2+^-binding proteoglycan family and binds with glycosaminoglycans to form part of the extracellular matrix (ECM) [25]. Cross-talk with ECM proteins exists throughout development and the misexpression of matrix molecules is a cause of many developmental defects [26]. The tumor microenvironment plays an essential role in cell proliferation and differentiation, and is key to the invasive and migratory capacity of tumor cells. A splice variant of the *SPOCK3* gene has been found to inhibit the expression of matrix metalloproteinase 2 (MMP-2), which is mediated by membrane-type (MT)-MMPs [27]. Activated MMP-2 was observed in various tumor tissues, suggestive of pro-MMP-2 activator(s) present in the tumor microenvironment [28,29]. MT1-MMP, an activator of pro-MMP-2 expressed on the surface of tumor cells, closely correlates with the invasive phenotype of human tumors [30,31]. All members of the SPOCK family, with the exception of *SPOCK2*, interfere with pro-MMP-2 activation mediated by MT1-MMP or MT3-MMP [27]. *SPOCK2* abolishes the inactivation of MT-MMPs by other *SPOCK* family members, causing ECM remodeling and allowing the migration of glioma cells expressing MT1-MMP [32]. Therefore, it is tempting to speculate that the expression of *SPOCK2* results in changes of the microenvironment that facilitate tumor cell migration. How this would translate in the specific passage through the BBB of the tumor cells remains to be elucidated.

Based on our previous RNA expression data, *GJD3* (also termed connexin 30.2 or Cx30.2) was identified as the most significant gene (*p* = 0.0002, permutation *p*-value ≤ 0.001, Figure 1) out of the 55 significantly differentially expressed genes, albeit with a narrow FOC between BM+ and BM− groups. Further analysis on the GJD3 protein status in 30 additional breast cancer samples showed a higher trend of GJD3 protein overexpression in the BM+ group. More than 50 percent of the BM+ breast cancer samples presented higher mean intensity values than those detected in the BM− samples (Figure 4e,f). Unfortunately, validation of *GJD3* at the mRNA expression level was not successful. This may be due to technical limitations: RNA from FFPE tissues may miss out in quantitative RT-PCR assays [33]. In addition, there may be significantly different median Ct values between RNAs derived from fresh-frozen samples, as compared to FFPE samples [34]. This is further supported by the significantly overexpressed *GJD3* mRNA levels in the fresh-frozen breast cancer samples that metastasize to the brain in the independent online cohort of 204 primary breast cancers (Appendix A). GJD3 is a protein member of the large family of connexins apt to form functional intercellular channels linking the cytoplasm of two adjacent cells and is localized to the plasma membrane, in areas of cell–cell contact [35,36]. Connexins are the sole proteins required for the assembly of gap-junctions [37]; this way of communication is thought to be essential to coordinate tissue growth, development, and physiological activities [36,38]. Differential regulation of connexin isotypes is fairly common and, in some cancers, connexins may display tumor suppressor potential [39] or oncogenic activity [40,41]. Dysfunctional connexins affect the growth control of cells, resulting in the rise of tumors [42]. Although deregulation of connexins and their gap-junctions has been implicated in breast carcinogenesis and tumor progression, specific correlations with tumor progression have only incidentally been reported [43,44], restricted to connexin isotypes from sub-family groups I [45,46], II [47], and III [48]. *GJD3* was never before associated with breast cancer and it was believed that ancient connexins (groups IV and V) are not expressed in cancers in the first place [42]. The role of *GJD3* expression in the formation of cerebral seeding of ER- breast cancer requires, at this point, further ratification.

Only a few patients developed brain metastasis shortly after the diagnosis of the primary tumor (2/10 in <12 months or 3/10 in <24 months) (Table 1). Therefore, the association of high expression of one or more of the three identified genes with early metastatic spread to the brain could not be calculated. The recent guidelines to treat patients with breast cancer include the use of neoadjuvant treatments. As a result, the availability of treatment-naïve primary breast cancer samples is very limited. For future aspects, it is important to investigate the effect of the various types of neoadjuvant treatments on the expression of *BOC*, *SPOCK2,* and *GJD3.* In addition, identifying genes involved with brain metastasis of patients who received neoadjuvant treatment is still needed.

## 5. Conclusions

The ability to identify breast cancer patients at risk for developing brain metastases may lead to new prophylactic intervention that will diminish morbidity and mortality. Here, we found the genes *BOC, SPOCK2,* and *GJD3* to be associated with metastatic potential of ER- breast cancer. These genes warrant further analysis as predictive biomarkers for the risk of brain metastases in patients with ER- breast cancers, and as targets for preventive strategies.

## Figures and Tables

**Figure 1 cancers-13-02982-f001:**
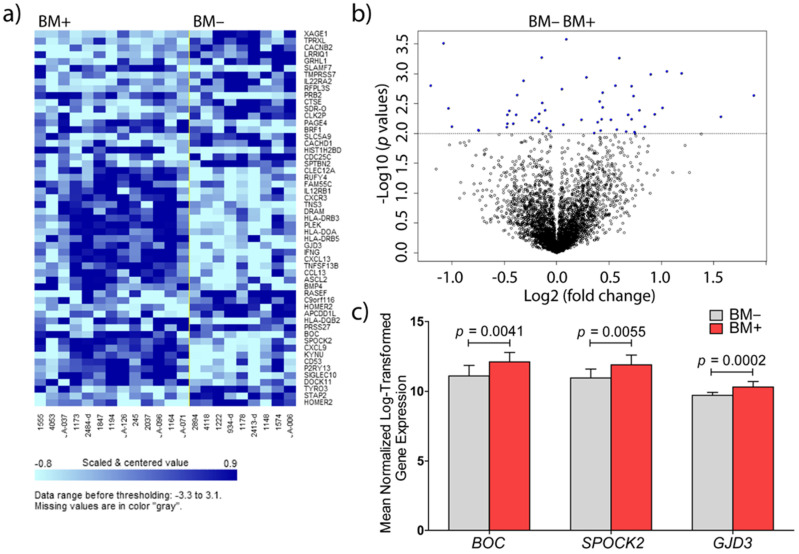
Differentially expressed genes in primary breast cancer samples with or without brain metastasis. (**a**) Clustered heatmap of the 55 significantly expressed genes (*p* < 0,01, univariate test) in fresh-frozen primary human breast cancer samples associated with the development of brain metastases (left) and breast cancer samples associated with metastasis to other organs (right); (**b**) volcano plot representation of the significantly expressed genes (*p* < 0,01) clustered in (**a**); (**c**) mean normalized Log 2-transformed *BOC, SPOCK2,* and *GJD3* differential expression between BM+ and BM− groups from the discovery sample set.

**Figure 2 cancers-13-02982-f002:**
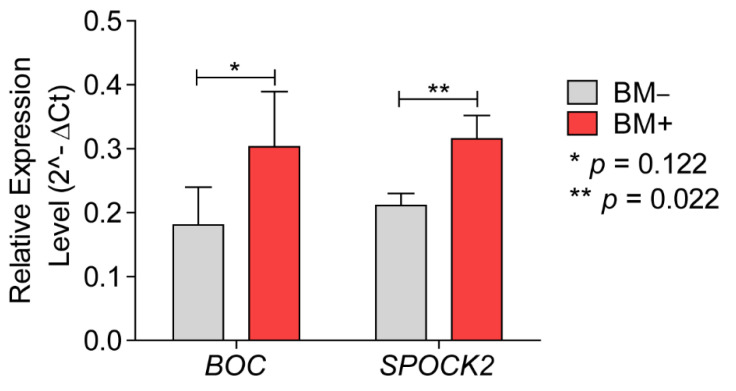
Relative mRNA expression of *BOC* and *SPOCK2* in primary breast cancers of patients who developed metastasis to organs excluding brain (BM−) and including brain (BM+). Bars indicate Mean ± SD.

**Figure 3 cancers-13-02982-f003:**
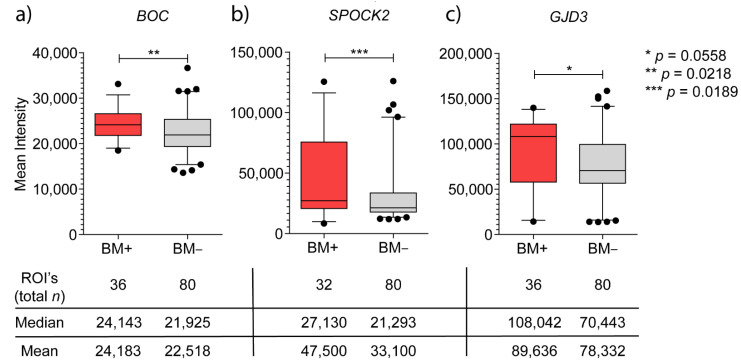
Box-whisker plots of (**a**) *BOC*-, (**b**) *SPOCK2*-, and (**c**) *GJD3*-IHC mean intensities, with tabular statistics summary for both BM+ and BM− groups of ER- primary breast cancers. The lowest and highest boundaries of the box represent the 25th and 75th percentiles, respectively. The solid line across the box indicates the median value. Error bars indicate the 5th–95th percentile.

**Figure 4 cancers-13-02982-f004:**
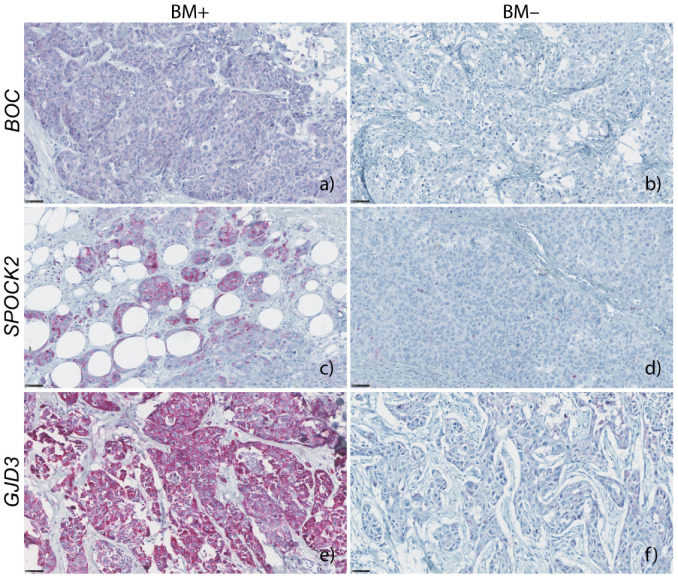
Representative *BOC*, *SPOCK2*, and *GJD3* positive and negative stainings of FFPE primary breast cancer samples. On the left column (**a**,**c**,**e**) are representative positive stainings from BM+ group; on the right column (**b**,**d**,**f**) are representative negative stainings from BM− group. Scale bars = 50 µm.

**Table 1 cancers-13-02982-t001:** Relevant clinical information regarding 30 primary breast cancer samples with systemic metastases from the validation cohort set. The clinical information provided includes age at diagnosis, estrogen receptor (ER), progesterone receptor (PR), and Her2/neu status of the primary tumor, lymph node status, neoadjuvant therapy, therapy post-breast cancer surgery (adjuvant therapy), the period from breast cancer diagnosis to detection of the first metastatic site (metastasis-free period in months), the period from breast cancer diagnosis to detection of brain metastasis (time to BM), and 1st and other metastatic sites. 10/32 primary breast cancers from patients who developed brain metastasis are underlined.

#	Age at Diagnosis (Years)	ER	PR	Her 2/neu	Lymph Node Status	Neoadjuvant Therapy	Adjuvant Therapy	Metastasis Free-Period (Months)	Time to BM (Months)	1st Metastatic Site	Other Metastatic Sites
1	32	neg	neg	neg	pos	-	CT + RT	6	11	liver	lung, brain
2	47	neg	neg	neg	pos	-	CT + RT	10	-	skin	-
3	53	neg	neg	neg	pos	-	CT + RT	48	-	skin	-
4	44	neg	neg	neg	pos	-	CT + RT	33	-	lung	-
5	42	neg	neg	neg	pos	-	CT + RT	24	-	liver	-
6	53	neg	neg	neg	pos	-	CT + RT	42	48	skin	brain
7	52	neg	neg	neg	pos	-	CT + RT	48	34	lung	brain, lung
8	59	neg	neg	neg	pos	-	CT + RT	35	-	pleura	bone, liver, lung, meninges
9	42	neg	neg	neg	pos	-	CT + RT	84	-	bone	liver
10	39	neg	neg	neg	pos	-	CT + RT	21	19	skin	brain
11	55	neg	neg	neg	neg	-	CT + RT	14	-	liver	bone
12	44	neg	neg	neg	pos	-	CT + RT	14	11	brain	meninges, pleura
13	61	neg	neg	neg	pos	-	CT + RT	26	56	bone, lung	liver, brain
14	64	neg	neg	neg	pos	-	CT + RT	7	-	lung	liver, bone
15	54	neg	neg	neg	pos	-	CT + RT	17	-	liver	skin, bone, leptomengieal
16	37	neg	neg	neg	neg	-	CT + RT	18	-	lung, liver, bone	
17	51	neg	neg	neg	neg	-	CT + RT	15	-	bone	lung, skin, liver
18	49	neg	neg	neg	neg	-	CT + RT	24	-	bone	lung
19	61	neg	neg	neg	neg	-	CT + RT	24	37	lung	brain
20	39	neg	neg	pos	neg	-	CT	51	95	lung	brain, bone
21	33	neg	neg	neg	pos	-	CT + RT	14	-	lung,	liver, adrenal
22	46	neg	neg	neg	pos	-	CT + RT	17	-	bone	bone, lung, liver
23	42	neg	neg	neg	pos	-	CT	51	-	lung	liver
24	63	neg	neg	neg	pos	-	CT + RT	42	-	bone	-
25	40	neg	neg	neg	neg	-	CT + RT	4	66	lung	brain
26	34	neg	neg	neg	pos	-	CT + RT	9	-	liver	bone
27	32	neg	neg	neg	pos	N.A.	N.A.	12	-	skin	-
28	25	neg	neg	-	neg	-	CT + RT	115	130	bone	liver, brain
29	53	neg	neg	neg	pos	-	CT + RT	15	-	pleura	-
30	33	neg	neg	neg	pos	-	CT + RT	34	-	lung	liver, adrenal

Neg = negative; pos = positive; N.A.—information not available; CT = chemotherapy; RT = radiotherapy.

## Data Availability

Dataset information is publicly available under GEO accession number GSE12276, EXP00013 [8]. The data presented in this study are available on request from the corresponding author.

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
