# Peer review of "Differential Expression of BOC, SPOCK2, and GJD3 Is Associated with Brain Metastasis of ER-Negative Breast Cancers"

_cancers, 2021, doi:10.3390/cancers13122982_

Round 1
Reviewer 1 Report
This manuscript by Pedrosa et al. describe that three genes BOC, SPOCK2, and GJD3 were overexpressed in the group of primary breast cancers which developed brain metastasis d by RT-PCR and by immunohistochemistry using an independent cohort of samples. Authors provided a conclusion that the overexpression of BOC, SPOCK2, and GJD3 is associated with the invasion of breast cancer into the brain. Although the manuscript was carried out carefully and has completed the writing describes the techniques and findings clearly. However, the data and innovation in the manuscript are not enough for publication in the journal. Further research confirmation is necessary.Author Response
We thank the reviewer for the comments. We agree that future studies should be carried out. We stated that in the manuscript (page 1, lines 35-37).
Reviewer 2 Report
This manuscript by Pedrosa et al. reports on the BOC, SPOCK2 and GJD3 genes that are associated with the brain metastasis in estrogen negative breast cancer. For both BOC and SPOCK2, the authors confirmed data at protein and mRNA levels in an independent breast cancer cohort, for SPOCK2 and GJD3 genes – in external online database. In summary, the manuscript is very interesting; however, I have some major points that I think the authors need to address.
Materials and Methods
- To validate the genes involved in brain metastasis of primary breast cancer, the authors use a cohort of 32 primary ER negative breast cancer samples. However, according to Table 1, sample 29 and 30 have ER positive cancer. This group is heterogeneous.
- Were both study groups and validation cohort matched for the treatment status and chemotherapy regimens? A more detailed description of the clinical data of validation cohort as well as data of online database cohort should be included.
Results
- Result sections are numbered inconsistently. Section 3.1 is followed by 2.2.
- Please, explain the lines 143-145.
- The authors show that BOC mRNA expression levels tend to be higher in the BM+ group. It will be better to include P-value in the Figure 2 and Suppl. Figure 1.
- Table 2 repeats Figure 3.
- To prove the specific action of study genes, it is need to validate on an own samples group that metastasize only to the brain, without other sites of distant metastasis (lung, liver, bone).
Reference
References 6 and 7 are the same.
Author Response
Materials and Methods
- To validate the genes involved in brain metastasis of primary breast cancer, the authors use a cohort of 32 primary ER negative breast cancer samples. However, according to Table 1, sample 29 and 30 have ER positive cancer. This group is heterogeneous.
We thank the reviewer for this valid point. We re-analyzed the validation cohort n=30 without the two ER+ samples. The results for BOC and SPOCK2 remained significantly different. The results for GJD3-IHC (Figure 3) were accordingly adjusted. We adjusted table 1 (samples 31 and 32 are now 29 and 30, respectively, pages 2 and 3), Figure 2 (page 6, line 182), and Figure 3 (page 8, line 209). Further adjustments in the text are specified at the end of this document.
- Were both study groups and validation cohort matched for the treatment status and chemotherapy regimens? A more detailed description of the clinical data of validation cohort as well as data of online database cohort should be included.
We thank the reviewer for this point. Both discovery and validation cohorts were matched to the best of our ability. 29/30 patients were confirmed not to have had any therapy prior to the breast cancer surgery (“Neoadjuvant Therapy” column has been added to Table 1 for clarification). Therapy indicated in Table 1 (now changed into “Adjuvant Therapy”) regards therapy post-breast cancer surgery. We added this explanation to the legend of Table 1).
As for the online database, there is no additional clinical data to refer to publicly. However, we believe that the clinical data we provided are sufficient to validate our findings. In case the reviewer would insist, we can provide the overview table as drafted below.
|
age |
Mean 51.09 (95% CI 49.28-52.91) |
||||||
|
|
|
n |
% |
||||
|
Age-cat |
<=40 |
46 |
22.55 |
||||
|
|
>40-<=55 |
86 |
42.16 |
||||
|
|
>55-<=70 |
55 |
26.96 |
||||
|
|
>70 |
17 |
8.33 |
||||
|
|
|
||||||
|
T |
T1 |
61 |
29.9 |
||||
|
|
T2 |
111 |
54.41 |
||||
|
|
T3/T4 |
26 |
12.75 |
||||
|
|
NA |
6 |
2.94 |
||||
|
|
|
||||||
|
N |
Neg |
107 |
52.45 |
||||
|
|
Pos |
97 |
47.55 |
||||
|
|
|
||||||
|
M |
M0 |
196 |
96.08 |
||||
|
|
M1 |
8 |
3.92 |
||||
|
|
|
||||||
|
Adj.Therapy |
yes |
73 |
35.78 |
||||
|
|
no |
123 |
60.29 |
||||
|
|
not app |
8 |
3.92 |
||||
|
Adjuvant therapy can be either hormonal or chemotherapy or a combination. |
|||||||
|
NA: not available |
|||||||
|
not app: not applicable (M1 cases) |
|||||||
Results
- Result sections are numbered inconsistently. Section 3.1 is followed by 2.2.
Previous Results section 2.2 corrected into 3.2 (line 163). Following sections also corrected: 2.2.1 into 3.2.1 (line 172); 2.2.2. into 3.2.2. (line 223); and 2.2.3. into 3.2.3. (line 239).
- Please, explain the lines 143-145.
Lines 143-145 do not belong to the result section of the manuscript and have been removed.
- The authors show that BOC mRNA expression levels tend to be higher in the BM+ group. It will be better to include P-value in the Figure 2 and Suppl. Figure 1.
We fully agree with the reviewer to add the p-value to significant differences. Our results indicate that BOC mRNA expression, although showing a trend, but was not significant. Therefore, ‘ns’ was annotated in the previous Figure.
However, to apply the reviewer request, the specific p-value was added to Figure 2. Page 6, line 182 and Supplementary Figure 1, page 7, line 190.
- Table 2 repeats Figure 3.
We agree, we created the table for extra clarification. However, we now combined Table 2 to Figure 3 to make it easier to follow (page 8, line 209).
- To prove the specific action of study genes, it is need to validate on an own samples group that metastasize only to the brain, without other sites of distant metastasis (lung, liver, bone).
We thank the reviewer for this suggestion. However, the discovery cohort included only two samples of patients who developed only brain metastasis (without other metastatic sites), which makes it impossible to perform any statistical test.
Reference
References 6 and 7 are the same.
We apologize. References corrected. All subsequent literature references throughout the manuscript have been accordingly adjusted.
Reviewer 3 Report
Three genes have been identified, which might be associated with formation of brain metastasis in patients with breast cancer. This is a significant topic, since management of brain metastasis is challenging and biomarkers for prediction of their development are lacking. There are some limitations to the manuscript which require further clarification:
Comment 1: There are some limitations to table 1. Therapy previous to collection of tissue samples should be outlined in more detail, since this represents a significant source of bias. Has there be treatment with immunotherapy or TKI? What does metastasis-free period exactly mean? The most interesting information would be the time from histopathological evaluation of the primary tumor to the detection of BM.
Comment 2: It should be outlined clearer, which tissue was investigate, and at which point in time it was collected.
Comment 3: Expression of the respective genes should be evaluated in matching pairs of primary tumor and BM tissue from the same patient. A retained molecular signature in both tissues would support the hypothesis that these genes might be involved in metastatic spread to the brain.
Minor comment:
The text in line 143-145: “This section….. can be drawn” seems to be not part of the manuscript.
Author Response
Comment 1: There are some limitations to table 1. Therapy previous to collection of tissue samples should be outlined in more detail, since this represents a significant source of bias. Has there be treatment with immunotherapy or TKI?
We thank the reviewer for this point. However, the specific chemo- and radiotherapy patients received varied within the group. The validation set is too small to discover any effects of particular agents or radiation dosages or application modalities. At this point, we do not see any additional value in detailing the specific types of chemotherapy. If the reviewer disagrees, we will try to retrieve the requested information. Therapy indicated in Table 1 regards therapy post breast cancer surgery and has now been changed into “Adjuvant therapy”. Further clarification has been given in the legend of Table 1.
None of the patients were treated with immunotherapy or with TKI.
What does metastasis-free period exactly mean? The most interesting information would be the time from histopathological evaluation of the primary tumor to the detection of BM.
the metastasis free-period relates to the period in between breast cancer diagnosis and the detection of the first metastatic site. We added the definition to the legend of Table 1 (page 3, lines 87 and 88). Histopathological evaluation was done immediately after the surgeries.
Comment 2: It should be outlined clearer, which tissue was investigate, and at which point in time it was collected.
For the discovery and validation cohorts, we used primary breast cancer samples. We have clarified that in the material and methods, section 2.1 (page 2, line 69). The tissue was collected immediately after surgery.
Comment 3: Expression of the respective genes should be evaluated in matching pairs of primary tumor and BM tissue from the same patient. A retained molecular signature in both tissues would support the hypothesis that these genes might be involved in metastatic spread to the brain.
We thank the reviewer for this interesting point. However, we did not study the metastatic tissue samples. Comparison of the molecular make-up of the primary tumors with that of the cerebral metastases was out of the scope of this study.
Minor comment:
The text in line 143-145: “This section….. can be drawn” seems to be not part of the manuscript.
We thank the reviewer. We correct this mistake and the text was deleted.
Round 2
Reviewer 1 Report
no comment
Author Response
We didn't find any comment to answer.
Reviewer 3 Report
The update has improved the manuscript, however there are still some limitations which require further clarification.
Comment 1: In Table 1, the time from diagnosis of cancer to the detection of BM should be supplemented. Is there a association of high expression of one of the genes with early metastatic spread to the brain?
Comment 2: The pathophysiological background how these genes might contribute to brain metastasis formation is still insufficiently investigated. Assessment of a differential expression of the three genes between the primary tumor and BM should be supplemented for those patients, in which tissue from neurosurgery is available.
Author Response
The update has improved the manuscript, however there are still some limitations which require further clarification.
Comment 1: In Table 1, the time from diagnosis of cancer to the detection of BM should be supplemented. Is there a association of high expression of one of the genes with early metastatic spread to the brain?
The time from diagnosis of breast cancer to the detection of BM has been added to Table 1, in column “Time to BM”, with corresponding legends. Only few patients developed brain metastasis shortly after the diagnosis of the primary tumor (2/10 in < 12 months or 3/10 in < 24 months). Therefore, the association that was requested by the reviewer can not be calculated.
Comment 2: The pathophysiological background how these genes might contribute to brain metastasis formation is still insufficiently investigated. Assessment of a differential expression of the three genes between the primary tumor and BM should be supplemented for those patients, in which tissue from neurosurgery is available.
We agree with the reviewer that the pathophysiology background is very important. However, this request is out of the scope of this work. The genes studied in this work were identified by comparing primary breast cancers with and without brain metastases. The genes were validated in the primary tumors of the discovery set and in an independent set of primary breast cancers of which part developed brain metastases. Comparisons of expression between the primary tumors and the brain metastases are out of the scope of this study. Moreover, we do not have a set of primary tumors and their brain metastases. In addition, the high expression of the genes in primary tumors does not necessarily reflect the expression in the brain metastasis. Genes are important to facilitate developing brain metastasis should not be the same genes that are important for the tumor to survive and proliferate at the metastasis site.
Round 3
Reviewer 3 Report
It is unfortunate, that no further sub-investigations can be made due to the low number of patients and missing samples. A new section in the discussion should better highlight these limitations.
Author Response
Dear reviewer,
We added a paragraph addressing the point of the reviewer to the discussion, line: 322-330.
"Only a few patients developed brain metastasis shortly after the diagnosis of the primary tumor (2/10 in < 12 months or 3/10 in < 24 months) (Table 1). Therefore, the association of high expression of one or more of the three identified genes with early metastatic spread to the brain could not be calculated. The recent guidelines to treat patients with breast cancer include the use of neoadjuvant treatments. As a result, the availability of treatment-naïve primary breast cancer samples is very limited. For future aspects, it is important to investigate the effect of the various types of neoadjuvant treatments on the expression of BOC, SPOCK2, and GJD3. In addition, identifying genes involved with brain metastasis of patients who received neoadjuvant treatment is still needed."